# Deep model predictive control of gene expression in thousands of single cells

**Jean-Baptiste Lugagne** ◉ [1,2] ✉, **Caroline M. Blassick** ◉ [1,2] & **Mary J. Dunlop** ◉ [1,2] ✉

Gene expression is inherently dynamic, due to complex regulation and stochastic biochemical events. However, the effects of these dynamics on cell phenotypes can be difficult to determine. Researchers have historically been limited to passive observations of natural dynamics, which can preclude studies of elusive and noisy cellular events where large amounts of data are required to reveal statistically significant effects. Here, using recent advances in the fields of machine learning and control theory, we train a deep neural network to accurately predict the response of an optogenetic system in *Escherichia coli* cells. We then use the network in a deep model predictive control framework to impose arbitrary and cell-specific gene expression dynamics on thousands of single cells in real time, applying the framework to generate complex time-varying patterns. We also showcase the framework's ability to link expression patterns to dynamic functional outcomes by controlling expression of the *tetA* antibiotic resistance gene. This study highlights how deep learning-enabled feedback control can be used to tailor distributions of gene expression dynamics with high accuracy and throughput without expert knowledge of the biological system.

Differences in gene expression dynamics can generate diverse cell phenotypes in genetically identical populations. These fluctuations can encode signals, provide temporal organization, and diversify communities, providing populations with the flexibility required to respond and adapt to environmental changes and stresses. For example, recent studies have demonstrated that the dynamics of entry into stationary phase in bacteria influence the emergence of antibiotic-tolerant persister cells[1], fluctuations in transcription factors can underlie bet-hedging strategies[2,3], and stochastic transcriptional bursts impact plasticity and drug resistance of cancer cells[4]. These examples highlight the critical role that single-cell gene expression dynamics can play in cell function and survival.

Despite broad recognition of the prevalence and importance of single-cell expression dynamics[5,6], two issues preclude measurements linking dynamics to function. First, gene expression is highly variable between individual cells in a population and within single cells over time[7,8], making large amounts of dynamic data necessary to capture the full range of behaviors within a given population and draw

statistically significant conclusions. Single-cell resolution studies enabled by genomics, transcriptomics, and flow cytometry have revealed a wide diversity of cell states and can quantify expression data with high throughput[9–12]. But these measurements tend to be "snapshots" disconnected from the cells' temporal context. Thus, current approaches typically specialize in either dynamic measurements or throughput, whereas functional studies require both. Second, we lack appropriate tools to generate arbitrary gene expression dynamics at the single-cell level in order to establish the causal relationship between expression and a cellular phenotype. Approaches such as optogenetics can be used to drive gene expression in single cells and have made headway into linking expression dynamics to function[13–15]. However, stochasticity makes it difficult to generate precise dynamics or subtle differences in gene expression that may be biologically relevant, because cells can exhibit different expression patterns even when exposed to identical optogenetic inputs.

Recently, studies have begun to circumvent this second issue by imposing expression dynamics with single-cell feedback control

[1]Department of Biomedical Engineering, Boston University, Boston, Massachusetts 02215, USA. [2]Biological Design Center, Boston University, Boston, Massachusetts 02215, USA. ✉e-mail: jlugagne@bu.edu; mjdunlop@bu.edu

platforms. With this approach, a gene of interest is made externally inducible, for example by using an optogenetic system, and its expression level is measured every few minutes via fluorescence microscopy. These data are processed on-the-fly by a control algorithm that decides whether to activate or repress expression of the gene in order to drive it towards a desired dynamic objective. Then, light is applied to stimulate single cells independently. This process is repeated every few minutes in a real-time feedback loop.

Algorithms that have been used to control gene expression dynamics include traditional proportional-integral strategies[16,17] and bespoke designs to test synthetic circuits[18] or cell interactions[19]. However, to date only approaches based on model predictive control have been used to assign time-varying dynamics to single cells, resulting in high levels of control accuracy[20,21]. In this type of controller, several candidate optogenetic stimulation strategies are considered and a model, often based on ordinary differential equations, is used to predict how the cell will respond to each strategy. Based on these predictions, the stimulation strategy that is expected to bring the expression level closest to a desired objective is applied to the cell.

However, building the mathematical models requires expert knowledge of the system, such as values for reaction rates and insight into which model structure is well-suited to the system, which can defeat the purpose of the approach. More importantly, the underlying

prediction models can be computationally expensive, especially if they integrate the complex dynamics that are a hallmark of stochastic gene expression. Because of this, studies using model predictive control have been limited to driving the dynamics of a few dozen cells in parallel, fundamentally hindering researchers' ability to identify intricate statistical relationships. Consequently, the potential of model predictive control in advancing our understanding of complex biological systems remains largely unrealized.

Recently, major milestones in control engineering outside of the field of biology have been achieved by algorithms that couple traditional control theory with machine learning[22]. One such recent approach is deep model predictive control[23,24], which uses deep neural networks to predict system responses to potential control strategies based on training data. These models have shown impressive accuracy at predicting the behavior of nonlinear and chaotic systems[25,26]. They can also incorporate high-dimensional data[25], and because neural network computations can be massively parallelized, they are orders of magnitude faster than traditional prediction models such as those based on ordinary differential equations[27]. Finally, developing the prediction model requires no expert knowledge of the system, making the approach transferrable to entirely different systems.

In this study, we use deep model predictive control to break the current limitations of throughput and dynamic control for single-cell

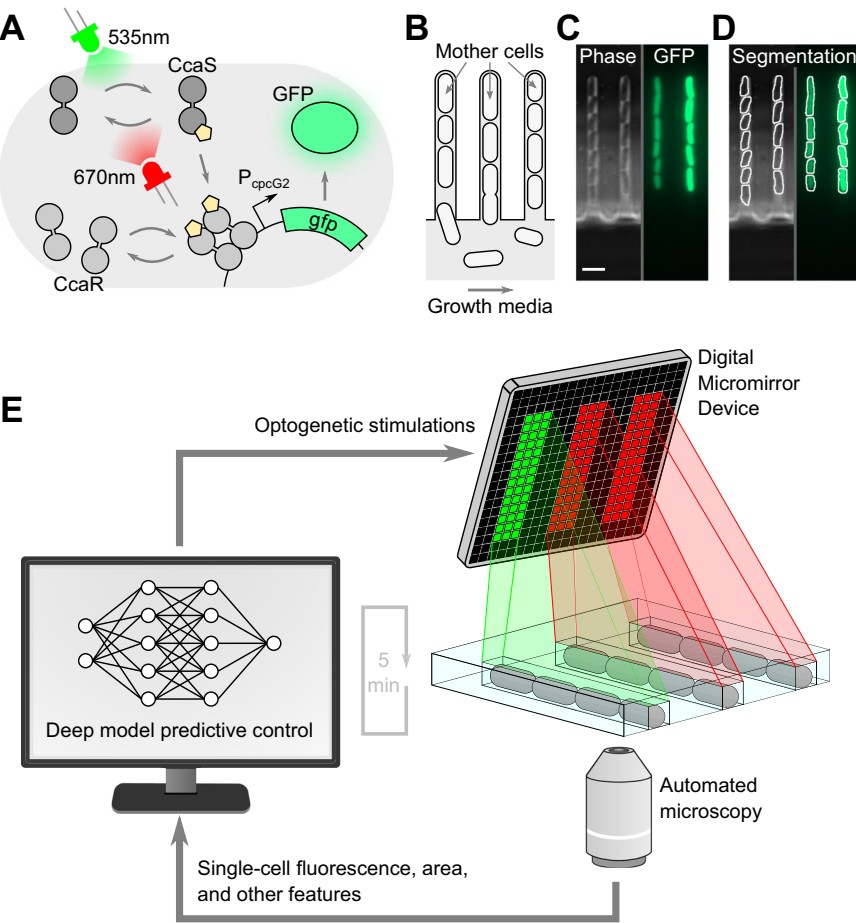

**Fig. 1 | Customized optogenetic stimulations can be applied to single *E. coli* cells in parallel. A** Schematic of the CcaSR optogenetic system. When exposed to green light, CcaS changes conformation and phosphorylates CcaR which then activates expression downstream of the P$_{cpcG2}$ promoter. Red light reverts the CcaS conformation, and CcaR unbinds from P$_{cpcG2}$. **B** Schematic of the mother machine microfluidic device. The mother cell is trapped at the end of the chamber while its progeny are pushed out into the channel where growth media flows. **C** Phase contrast and GFP microscopy images of cells growing in the mother machine. Scale bar, 4 μm. **D** DeLTA segmentation masks on images from (**C**). Segmentation is performed on phase contrast images, and morphological features as well as fluorescence are extracted based on these masks. **E** Schematic of our experimental platform. Every 5 min, automated microscopy and data analysis of cells growing in the mother machine are performed, and single-cell data are fed into a deep model predictive control algorithm. The algorithm decides which optogenetic stimulations to apply to each cell. Customized, chamber-specific light is applied accordingly via a Digital Micromirror Device.

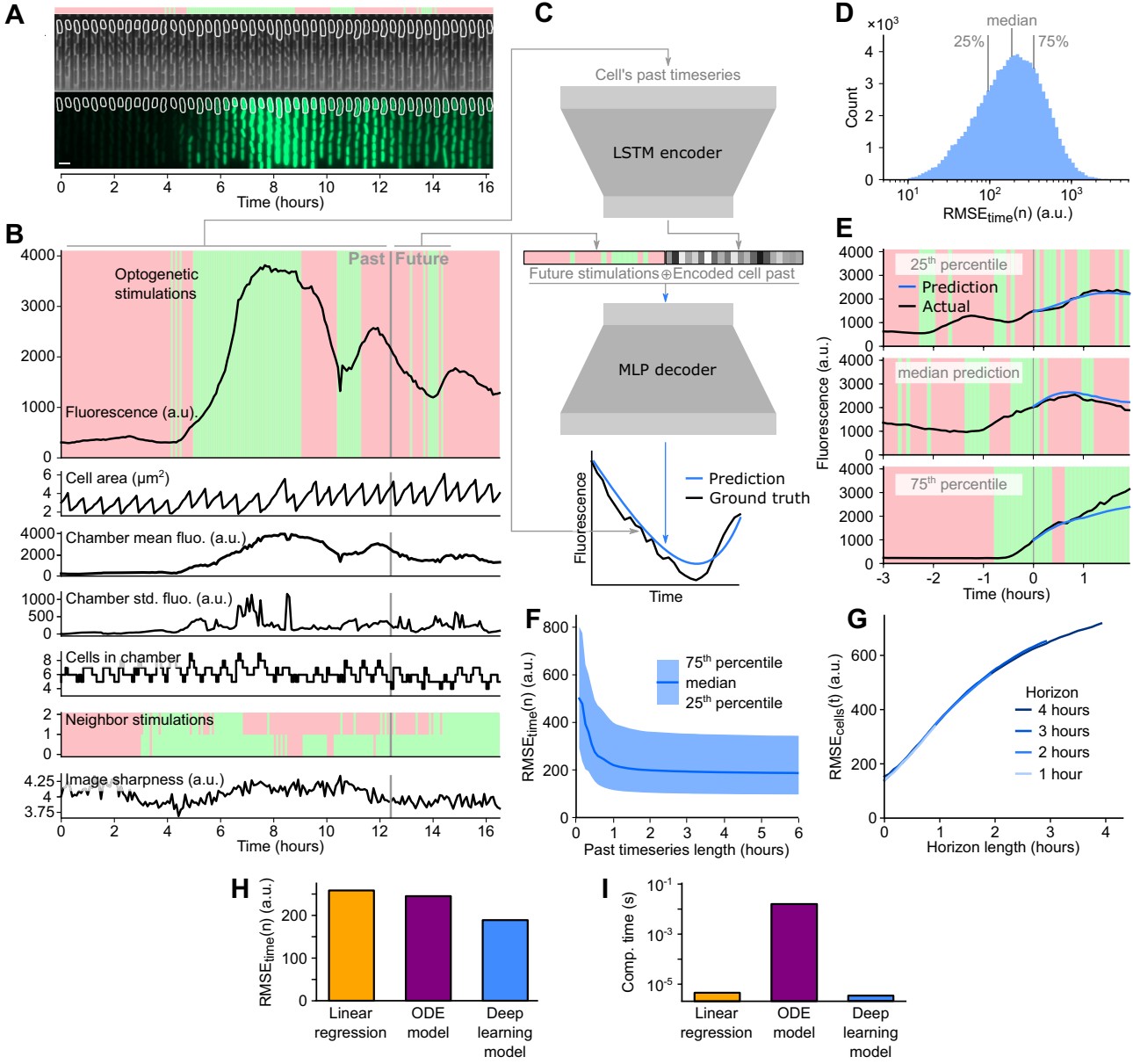

**Fig. 2 | An encoder-decoder deep neural network predicts single-cell responses to optogenetic inputs. A** Kymograph of a representative single cell subjected to a random stimulation sequence for training set generation. The red-green bar at the top shows the sequence of random optogenetic stimulations the cell was subjected to. White outline shows segmentation mask for the mother cell. Scale bar, 4 μm. **B** Timeseries of eight extracted features for the cell shown in (**A**). For timeseries forecasting, a randomly selected time point splits the timeseries into past and future, as indicated with a vertical gray line. **C** Schematic of the timeseries forecasting model. All past features of the timeseries are fed into a long short-term memory (LSTM) encoder, which encodes relevant information about the cell's past into a single 32-dimensional vector. This vector is then concatenated with a binary vector of the future optogenetic stimulations the cell was subjected to. The multilayer perceptron (MLP) decoder then predicts the response of the cell to the future light sequence, which is compared to the ground truth of the known future fluorescence trajectory. **D** Histogram of $RMSE_{time}(n)$ for 100,000 samples from the

validation dataset for the 2 h horizon model. The 25th percentile, median, and 75th percentile are marked with light gray lines. **E** Trained model predictions for the 2 h horizon model on the validation dataset. The red-green background represents the optogenetic stimulations that were applied to cells in the validation experiment. The black curve represents actual single-cell fluorescence. The vertical gray line shows the arbitrary partition of the validation timeseries into past and future. For illustration purposes we only show the past 3 h, but past timeseries can be longer. The blue curve represents the model's prediction of the cell's response. Sub-panels illustrate results in the 25th percentile, median, and 75th percentile prediction error. **F** $RMSE_{time}(n)$ for different lengths of past timeseries data. **G** $RMSE_{cells}(t)$ over validation dataset for different prediction horizon lengths. **H** Median $RMSE_{time}(n)$ for predictions from linear regression model, ODE-based model, and our deep learning model. **I** Computation time per sample for linear regression model, ODE model predictions, and our MLP decoder.

gene expression. We first develop a high-throughput experimental platform to grow, observe, and optogenetically stimulate *E. coli* bacteria at the single-cell level. We demonstrate that it is possible to predict gene expression in single cells with deep learning models with high accuracy. We then use these models to control dynamic gene expression in thousands of cells in parallel with a high degree of

precision, both at the population level and in single cells. Finally, we apply single-cell control to expression of an antibiotic resistance gene, producing high resolution data about the relationships between expression levels, growth rate, and survival. Overall, these findings demonstrate the power of deep model predictive control for driving single-cell gene expression dynamics in a range of experimental

contexts, including enacting arbitrary gene expression patterns, or tightly controlling expression levels in single cells.

## Results

### Experimental setup

Optogenetic systems can regulate gene expression via light, a signal that is easy to integrate with computational control. Here, we used the well-characterized CcaSR optogenetic system in *E. coli*[28,29] (Fig. 1A). When exposed to 535 nm green light, CcaS changes conformation to phosphorylate the CcaR transcription factor, which then binds to the $P_{cpcG2}$ promoter and activates the expression of downstream genes. 670 nm red light reverts CcaS back to a low-kinase conformation, effectively turning off gene expression. In our system, we put a gene encoding green fluorescent protein (*gfp*) downstream of $P_{cpcG2}$. In this way, green light stimulations activate green fluorescence expression in our cells, and red light stimulations repress it. To enable simultaneous control of thousands of cells in parallel, we grew *E. coli* in the mother machine microfluidic device[30], where cells are constrained to grow in single-file lines within thousands of short (25 μm), parallel chambers (Fig. 1B). Trapped "mother" cells at the dead end of each chamber can be observed for hours to days. As a mother cell grows and divides, it produces "daughter" cells, which are pushed down the chamber. In our experiments, we acquired phase contrast and green fluorescence images (Fig. 1C) every 5 minutes for ~150 different positions, where each field of view contains 27–28 mother machine chambers, for a total of ~4000 mother cells. We analyzed microscopy images in real time with the deep learning enabled time-lapse analysis software DeLTA[31,32], extracting cell features such as fluorescence levels on-the-fly, while the experiment was running (Fig. 1D).

In order to activate the optogenetic system, we exposed cells to green light pulses. Conversely, red light pulses were used to repress gene expression. To selectively illuminate individual chambers within the microfluidic chip we used a digital micromirror device (DMD). The DMD projects a user-defined image onto a sample, and we designed these illumination patterns to selectively stimulate specific chambers within the chip (Fig. 1E). The DMD thus forms the final component of our computer-based feedback loop. Automated measurements are acquired via time-lapse microscopy, processed with an image analysis pipeline, and fed into a deep model predictive control algorithm, which decides the customized optogenetic stimulations that the DMD will apply in order to drive each mother cell's fluorescence towards a desired objective.

### Gene expression forecasting

A crucial part of the deep model predictive control framework is the ability to accurately forecast the effect of light stimulation on future gene expression, such that the appropriate perturbation can be selected to produce the desired gene expression dynamics. To this end, we first asked whether it was possible to predict single-cell optogenetic responses using a deep learning model. We conducted experiments without feedback control to acquire four training and three validation sets, in total containing 15,898 (training) and 13,811 (validation) single-cell timeseries. In these experiments, we monitored the response of mother cells to randomized sequences of optogenetic stimulations for 16+ hours (Fig. 2A). We applied optogenetic stimulations, acquired phase contrast and fluorescence microscopy images, and extracted data in real-time every five minutes, corresponding to 192+ time points per cell. For each mother cell at each time point we recorded eight features (light stimulations, fluorescence, and cell area of the mother; mean and standard deviation of fluorescence for all cells in the chamber; number of cells in the chamber; stimulations applied to the neighboring chambers; and image sharpness) (Fig. 2B). It was not clear, a priori, which of these features would be relevant for predicting the response of the mother cell to optogenetic stimulations, but a key advantage of using deep learning models over

traditional biochemical models is their flexibility in handling multi-dimensional data, allowing the model to discern which features are most informative.

To predict cell responses, we built an encoder-decoder deep learning model (Fig. 2C). The encoder consists of two cascading long short-term memory (LSTM) networks[33] that encode the entire past trajectory of a single cell into a small latent space. The resulting 32-dimensional vector in this latent space is thus a fixed-size representation of the cell's entire past. The decoder is a multi-layer perceptron (MLP) that uses this encoded vector to predict the cell's fluorescence response for a candidate series of future optogenetic stimulations. Training samples were generated by randomly selecting cells in the training set and separating their timeseries into arbitrary past and future timepoints. The entire past timeseries together with the future optogenetic stimulations were compiled as training inputs, while the measured future fluorescence levels of the cell were used as ground truth for training. We trained the model over 500 epochs of 200 batches, with each batch comprising 100 training samples. Mean squared error between prediction and ground truth was used as training loss. We minimized this loss during training with the Adam algorithm[34] for gradient descent. Overall, the goal of our encoder-decoder model is to rapidly and accurately predict the response of a cell to a candidate set of light stimulations based on its past.

Following training, we evaluated the ability of our model to predict single-cell responses over a 2 h prediction horizon. Because in experiments we generate data both over time and over many cells, we use two main metrics to evaluate error: root mean square error computed across time as a function of cells or samples, $RMSE_{time}(n)$ and root mean square error computed across cells or samples as a function of time, $RMSE_{cells}(t)$ (Methods). These metrics differ in which axis is used to compute the error and provide alternative views of performance, where $RMSE_{time}(n)$ is used to view a distribution of errors while $RMSE_{cells}(t)$ provides a view of how error evolves with time. We compared 100,000 single cell predictions from the validation dataset to the corresponding ground truths and computed $RMSE_{time}(n)$ between them (Fig. 2D). We illustrate what these values represent in terms of single-cell predictions for 25th percentile, median, and 75th percentile of the error distribution in Fig. 2E. We also evaluated the impact of different hyperparameters for our deep learning model, such as the number of layers and the number of units per layer, on both prediction accuracy and inference time (Fig. S1). Unsurprisingly, we found that very small networks provide faster inference but poorer accuracy. However, we also found that after a certain size, larger networks also experience a decline in accuracy, likely due to over-fitting. The model hyperparameters we used for subsequent experiments balance this tradeoff and produce results with low errors. In this model, the encoder uses LSTM networks with 64 and 16 units and the decoder uses an MLP network with 5 layers of 32 units. Additionally, we evaluated how sensitive our approach is to changes in the training data by fitting the same network against different combinations of the same experimental datasets and found that prediction accuracy was similar regardless of the datasets used, indicating the robustness of our approach (Fig. S2). Overall, the ability of the deep learning model to predict GFP dynamics over a 2 h horizon is excellent, with the vast majority of model predictions showing good agreement with the ground truth. The most erroneous predictions (95th percentile) tend to be caused by hard to predict events, for example where the cell suddenly stops responding or there are glitches in image analysis (Fig. S3).

The speed of the deep learning model predictions is a critical factor for our deep model predictive control use case. Under experimental conditions, the encoder processes past timeseries in 800–2200 μs depending on timeseries duration, and the decoder takes 20 μs to predict fluorescence when samples are processed in parallel. In a single instance of the feedback loop, the encoder runs

once per cell while the decoder runs 1000 times per cell to test candidate stimulation strategies. This process is parallelized for ~4000 cells, typically totaling under 90 s to compute model predictions for all cells, which is well under the 5 min acquisition interval of the experiments. The trained model can predict expression trajectories with impressive accuracy and speed, even against non-trivial optogenetic sequences where activation and repression alternate frequently, a notable achievement given the stochastic nature of single-cell gene expression.

Next, we evaluated how gene expression prediction accuracy was impacted by the length of the past timeseries or the prediction horizon. The model is fed data from the past in order to predict the future over a defined horizon. Both values—the past timeseries length and the prediction horizon length—are likely to impact model performance. For past timeseries, the LSTM encoder can handle any amount of data, but we investigated the minimum amount needed for accurate predictions. The answer to this has implications for experiment durations and can indicate the extent to which "memory" of past events impacts future expression patterns. To test the impact of timeseries length, we fed artificially truncated past timeseries of varying length into the encoder (Fig. 2F). We found that after 1.5 h of past data the error plateaus, indicating that model performance is highly dependent on data from the recent past.

Next, we asked how prediction horizon length impacts prediction accuracy. While the horizon needs to be long enough to predict the long-term consequences of control actions and account for inherent delays due to transcription and translation, predictions typically become less accurate over longer timescales. We conducted independent trainings of the model to predict single-cell fluorescence over horizons of 1, 2, 3, and 4 h (Fig. 2G, Fig. S4). We evaluated $RMSE_{cells}(t)$ of the model predictions against the ground truth of the validation set. As expected, in all cases the error increases for time points further into the future. But interestingly, we found that when comparing across horizon lengths the error between the same time points in each horizon was indistinguishable, suggesting that horizon length is flexible and can easily be adapted to different use cases. For model predictive control, genetic systems with long delays or strong nonlinearities might benefit from longer prediction horizons, while shorter horizons mean fewer control strategies to evaluate and therefore lower computation requirements.

We also evaluated how the different measured features impact the ability of the network to predict future cell fluorescence. Although we measured eight features for each mother cell at each time point, it was not clear which of these features would be most informative for predicting a cell's future response. Thus, for each single-cell feature, we masked out that feature's data, retrained the network, and then evaluated prediction accuracy (Fig. S5). The past mother cell fluorescence stood out for its role in improving prediction accuracy, although ultimately we found the best accuracy overall when using a model trained on all features. For completeness we also evaluated a model where both mother cell fluorescence and mean fluorescence in the whole chamber were removed, which made the system completely blind to past fluorescence levels and led to a large decrease in accuracy, as well as a model where all cell features were removed (i.e. no past data was fed to the prediction model), which unsurprisingly caused prediction accuracy to collapse. For subsequent experiments, we continued to use the full range of all eight measured features to maximize prediction accuracy.

In addition, we investigated how past timeseries were encoded into latent space representation by looking at 2D embeddings of our validation dataset after it was processed by the encoder (Fig. S6). Since our data is not fundamentally categorical, there are no obvious clusters in those embedded spaces. However, manual inspection of points in these spaces reveals a general structure as well as co-localization of similar traces based on noisiness, errors, or response strength,

indicating that the latent space is an intricate encoding of not only cell behavior but sources of uncertainty and noise. For instance, traces from filamenting cells tend to appear together in the same region of the space.

Finally, we compared the accuracy and computation time of our deep learning approach to other classes of predictive models. First, we implemented a simple linear regression model to infer the fluorescence level at each future timepoint from past timeseries data (Supplementary Text). Prediction accuracy of the linear regression model was inferior to results with the deep learning model (Fig. 1H), producing a median error comparable to that of our deep learning model with the mother cell fluorescence feature masked out (Fig. S5). However, the linear model predictions were not unreasonable, dramatically outperforming the case where all features were masked out, suggesting that this simple model may produce tolerable predictions for some applications. To assess computation time, we compared the prediction time of the linear regression to the decoder part of the deep learning model. In a typical feedback loop iteration, the encoder is run once per cell while the decoder is run 1000 times, thus the decoder speed has a much higher impact on throughput than the encoder in our model predictive control framework. We found that the decoder slightly outperformed the linear regression model (Fig. 1I). The reason for this is that the input to the decoder is a small (32-dimensional) vector representation of the cell's entire past, whereas the input to the linear regression includes all timepoints and all features. The computation time associated with the encoder is slower than linear regression, but is run much less frequently (Fig. S7). To compare our deep learning model to the previous state of the art in single-cell control of gene expression, we re-implemented the approach described by Chait et al.[20] based on ordinary differential equations and hybrid Kalman filtering, and fitted model parameters to our data (Supplementary Text). We found that this approach outperformed the linear regression model on prediction accuracy, but did not reach the levels of accuracy achieved by our deep learning model (Fig. 2H). We suspect this is because of the underlying assumption in Kalman filters that noise is Gaussian, which is generally not the case for gene expression and leads to sub-optimal state estimation[35]. Critically, the differential equation approach was several orders of magnitude slower (more than 4000 times slower) than our deep learning approach (Fig. 2I). To make sure that our implementation was not artificially limiting the performance of the differential equation and Kalman filter based approach due to improper combinations of parameters, we also evaluated a broad range of technical parameters and analyzed performance in more detail for the three best combinations of parameters (Fig. S7, S8, Supplementary Text), finding consistent results for performance across all three. Taken together, these results demonstrate that our deep learning model not only allows us to make predictions that are highly accurate, but that rapid computation times also make it possible to drastically increase prediction throughput over the current state of the art. These benefits in accuracy and computation time join other advantages. For example, the deep learning model does not require mechanistic knowledge of the system's behavior, which is necessary to construct an ordinary differential equation model. For these reasons our deep learning model is particularly well suited to the task of single-cell model predictive control of gene expression, and offers the potential for capitalizing on the throughput permitted by automated single-cell microscopy.

## High-throughput single-cell control

Model predictive control algorithms use forecasting models to predict the response of the controlled system to a range of potential input strategies. The model prediction for each strategy is compared to the control objective, and the best strategy is identified and applied (Fig. 3A) and reevaluated regularly. In our case, real-time measurements of cell fluorescence and other features were used as inputs to

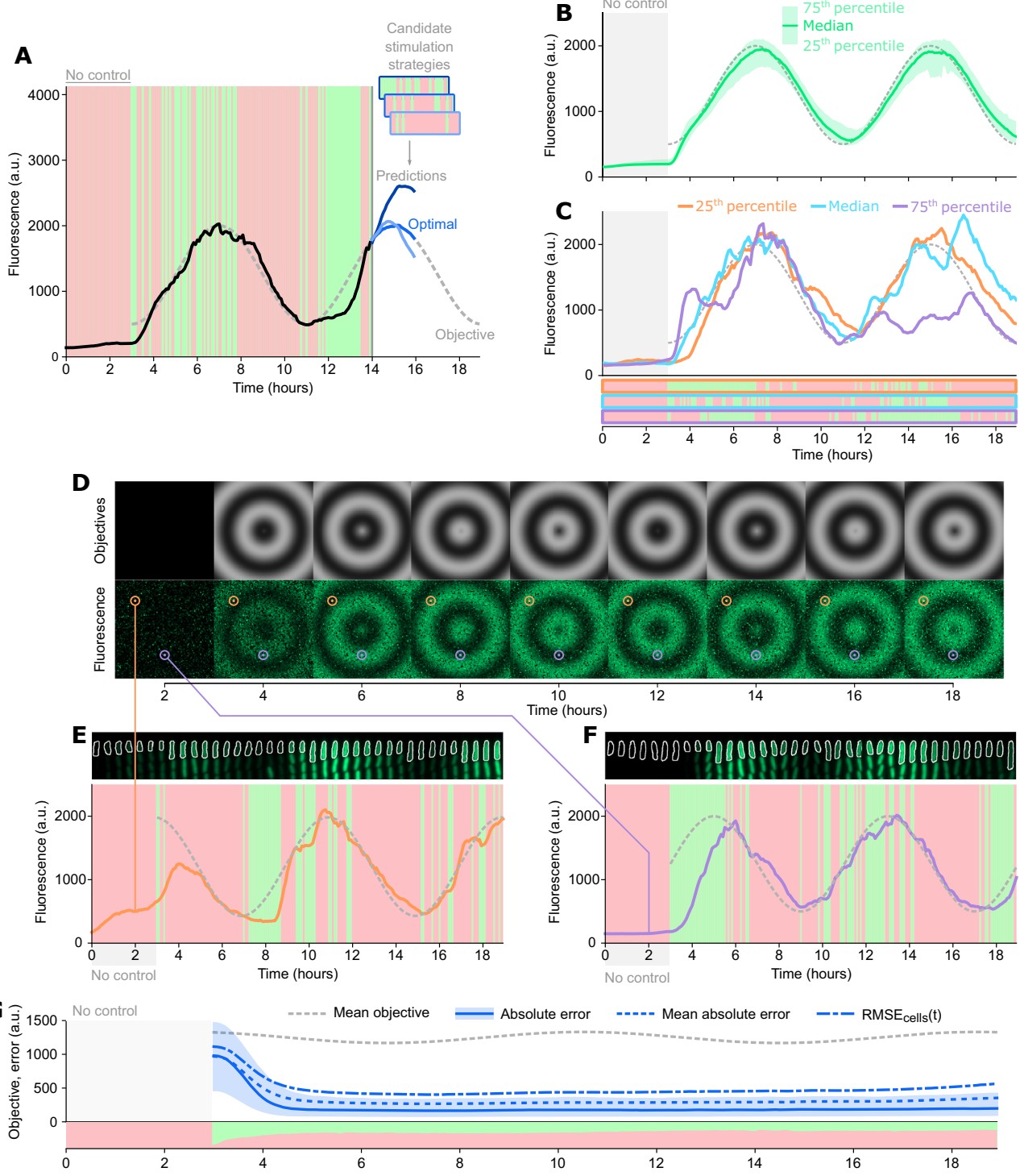

the gene expression forecasting model at every time point. These input data are combined with many potential control strategies, i.e., optogenetic stimulation sequences that could be applied over the prediction horizon, to produce cell response predictions. The stimulation strategy that is predicted to minimize the $RMSE_{time}(n)$ between the cell's fluorescence and a predetermined objective is selected, which is then implemented via the DMD. This entire process is repeated every five minutes, incorporating real-time information to improve control fidelity.

Because we apply a new light stimulation every five minutes (i.e., 12 per hour) and the model can choose between two options (red or green light), exhaustively testing all possible stimulation patterns for a horizon of length L hours requires $2^{12L}$ model runs per cell. For a L = 1 h horizon, this results in 4096 possible light application patterns to consider. For L = 4 h, exhaustively testing all options requires ~$3 \times 10^{14}$ predictions for each cell. Brute force evaluation of all control strategies therefore rapidly becomes computationally intractable. To circumvent this issue, we used a binary particle swarm optimizer[36]. We ran 40 particles, with each particle predicting the cell's response to a random optogenetic stimulation pattern and computing the error between the prediction and control objective. The best predicted outcomes for each particle, as well as the best outcome overall, were used to update

**Fig. 3 | Deep model predictive control can accurately drive time-varying expression dynamics in 10,000 independent cells. A** Model predictive control principles. The control algorithm predicts the cell's response to several potential control strategies, and selects the one closest to the control objective. Different shades of blue represent the predictions for different potential strategies. Three candidates are shown here for illustration, but 1000 are evaluated per cell and timepoint. **B** Control performance at the population level. The dashed curve represents the control objective. The solid green curve shows the median fluorescence of the population ($n = 524$ cells). The shaded area represents 25th to 75th percentiles of the population fluorescence. **C** Control performance for representative single cells. Colored solid curves represent single-cell fluorescence trajectories in the 25th percentile, median, and 75th percentile of control error. Below, red and green timeseries represent the optogenetic stimulation sequences that were applied by the controller for each single cell. **D** Expanding concentric sinewaves movie. The top row shows the control objectives that were assigned to each

cell in a $100 \times 100$ pixel movie. The bottom row shows measured fluorescence values for single cells subjected to deep model predictive control in the experiment ($n = 10,000$ cells). The orange and purple circles highlight the position of representative cell "pixels" shown in panels (**E, F**). **E** Example of control performance for a single cell at pixel coordinates (15, 15) in the expanding sinewaves movie. Fluorescence kymograph for the cell is shown at the top. Single cell response to control inputs is plotted at the bottom, with control objective shown as a dashed gray line. **F** Example for a single cell at pixel coordinates (75, 50). **G** Error metrics over time. The dashed gray line represents the average objective across cells, over time. The blue shaded area represents the 25th to 75th percentile of the absolute error, and the solid blue line represents the median absolute error. Dashed lines represent the mean absolute error and $RMSE_{cells}(t)$. The red and green shaded regions at the bottom represent the proportion of cells that were exposed to red or green optogenetic stimulations.

the probability that each bit in the optogenetic stimulation pattern was set to red or green. We then sampled and evaluated new random optogenetic sequences, iterating the whole process 25 times. This approach efficiently found optimal or near-optimal solutions with only these 1000 model predictions, even for the 4-hour horizon case (Fig. S9, Supplementary Text). Previous studies that have used traditional model predictive control to drive gene expression using differential equation and Kalman filter approaches have had shorter time horizons of 8 and 4 timepoints[20,21], corresponding to 256 and 16 total potential strategies to evaluate, which they used to control a few dozen cells. In comparison, our deep learning model can evaluate 1000 strategies each for thousands of cells, corresponding to millions of predictions every five minutes.

We next integrated the encoder-decoder deep learning model into the full feedback design to achieve real-time control of single cells. First, as a small test case we evaluated the performance of our 2 h horizon controller on ~500 single cells driven towards a sinewave objective. We initially exposed cells to only red light stimulations for 3 h for equilibration. We then turned on the deep model predictive control algorithm for t = 3 to t = 19 h. The median fluorescence of the entire population follows the objective well (Fig. 3B). More importantly, individual cells also stay close to the objective. We computed the single-cell $RMSE_{time}(n)$ between fluorescence and objective. Example single-cell trajectories of cells at the 25th percentile, median, and 75th percentile of error show the ability of the control system to track a dynamic objective function (Fig. 3C, Movie S1). The controllers adopt complex strategies, anticipating cell responses and adapting to each cell based on its past behavior. This is visible by comparing the light stimulation patterns across cells, which are customized to each cell despite identical objective functions, indicating the necessity of single-cell level control. During the same experiment, we also evaluated performance for the controllers with different horizon lengths (Fig. S10). For all models, control performance was very similar across horizons (Figs. S11, S12). This was slightly surprising, as we expected at least some variability in performance between the different algorithms, but can be explained by the fact that the genetic system we are controlling is fairly straightforward, and responds rapidly enough for the 1-hour control horizon to be sufficient. However, the ability to predict over longer horizons is likely to be useful in systems with long delays or complex dynamics. Overall, we found that all prediction horizons ranging between 1 and 4 h yielded extremely accurate single-cell resolution tracking of a dynamic objective function.

A key advantage of computer-based feedback is that each single cell can be assigned its own customized control objective, making it possible to induce complex dynamic phenotypes that are visible at the population level. To demonstrate the capabilities of our approach both in terms of accuracy and throughput, we used our control algorithm to produce movies where each pixel's intensity was the control objective for a single cell's fluorescence. First, we generated the

objective functions associated with a $100 \times 100$ pixel movie of concentric sinewaves expanding outwards (Fig. 3D, S13A, B, Movie S2). In practice, each cell was assigned a sinewave objective function like that of the previous experiment but with a different phase delay depending on its position in the $100 \times 100$ pixel array. We obtained these results in a series of three experiments, each controlling 3000–4000 cells, for a total of 10,000 cells. Hardware constraints prevented us from running this as a single experiment with 10,000 cells; however, we note that the control algorithm would be capable of handling this throughput. The reconstructed single-cell fluorescence movie is remarkably similar to the original, and single-cell trajectories follow their own objectives closely (Fig. 3E, F). We quantified control accuracy across all cells using several metrics. As expected, error decreases sharply in the first two hours as the system transitions from an equilibration period with red light only to tracking the objective (Fig. 3G). After this initial decrease, the error remains relatively constant. We found that the error correlates with the objective value, where higher objectives tend to lead to higher absolute error (Fig. S13C). Interestingly, this relationship is not absolute: Error is smallest for phases of the sinewaves that follow shortly after the minimum in objective value, and maximal for phases that slightly precede the peak objective value (Fig. S13D), indicating that the intrinsic dynamics of the circuit do not allow it to perfectly follow the dynamics of the objective at higher and lower values of fluorescence. The mean absolute error and $RMSE_{cells}(t)$ were all low following initial transients, with the proportion of cells receiving red or green light also remaining relatively constant through t = 19 h (Fig. 3G). Overall, this result demonstrates our ability to simultaneously control thousands of individual cells with customized objective functions with high accuracy.

Our algorithm is not limited to following sinewaves and can be used to track arbitrary objectives. To demonstrate this on a challenging objective function, we reproduced a scene from Stanley Kubrick's epic science fiction film *2001: A Space Odyssey* (Fig. 4A, Movie S3). For the scene, we used a resolution of $125 \times 80$ pixels, where the 10,000 individual objective functions were assigned to cells partitioned into three 32 h long experiments. We selected this clip because it is a complex scene with high dark-light contrast and contains both pixels that rapidly transition from dark to light or the reverse, and those where the objective function must hold a constant value for a long duration. Both the scale of the scene, which requires thousands of pixels, and the dynamic complexity of the movie clip, make this a challenging test case.

We found that the deep model predictive control algorithm was able to reproduce the clip with good accuracy, with the main subject of the scene easily identifiable and details of the foreground and background also visible (Fig. 4A). Some movements are too fast, and the cells cannot respond quickly enough to track sudden, sometimes fleeting changes in their control objectives (Fig. S14A). Subtle nuances in image contrast are also hard to reproduce perfectly, creating a

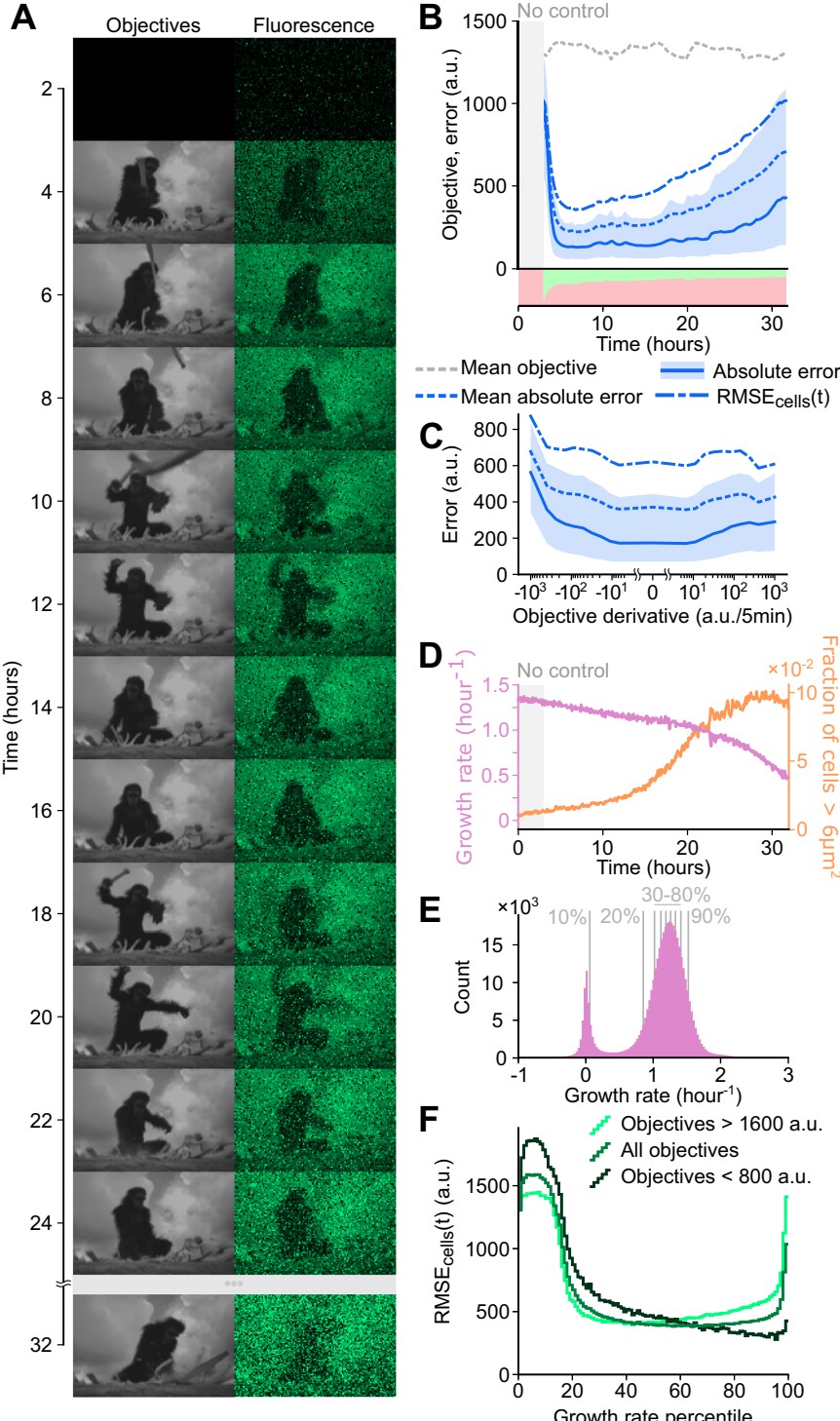

**Fig. 4 | Control performance against arbitrary, complex objectives is limited by cell growth and death. A** *2001: A Space Odyssey* scene reproduction. The first column shows the control objectives that were assigned to each cell. The second column shows measured fluorescence values for single cells subjected to deep model predictive control in the experiment ($n = 10,000$ cells). Image stills taken from *2001: A Space Odyssey,* reproduced with permission from Warner Brothers, all rights reserved. **B** Error metrics over time. The dashed gray line represents the average objective across cells, over time. The blue shaded area represents the 25th to 75th percentile of the absolute error, and the solid blue line represents the median absolute error. Dashed lines represent the mean absolute error and $RMSE_{cells}(t)$. The red and green shaded regions at the bottom represent the proportion of cells that were exposed to red or green optogenetic stimulations. **C** Error metrics as a function of the time-derivative of the objective. To show performance across a diverse range of time-derivatives, we use a log scale for objectives with both positive and negative derivatives. **D** Median cell growth rate across all cells over time (light blue). The fraction of cells larger than $6\ \mu m^2$ (orange) over time indicates the presence of unhealthy filamented cells. **E** Histogram of growth rates, over all cells and timepoints. Single-cell growth rate trajectories were smoothed with a median filter sliding over a one-hour time window. Percentiles of the distribution are shown with vertical gray lines. **F** $RMSE_{cells}(t)$ per percentile of growth rate. Error is computed for cells and timepoints belonging to each single percentile from the growth rate distribution shown in (**E**). Light green corresponds to objectives with a value below 800 arbitrary units of fluorescence, while dark green represents objectives above 1600 units of fluorescence. Further error analysis is provided in Fig. S14.

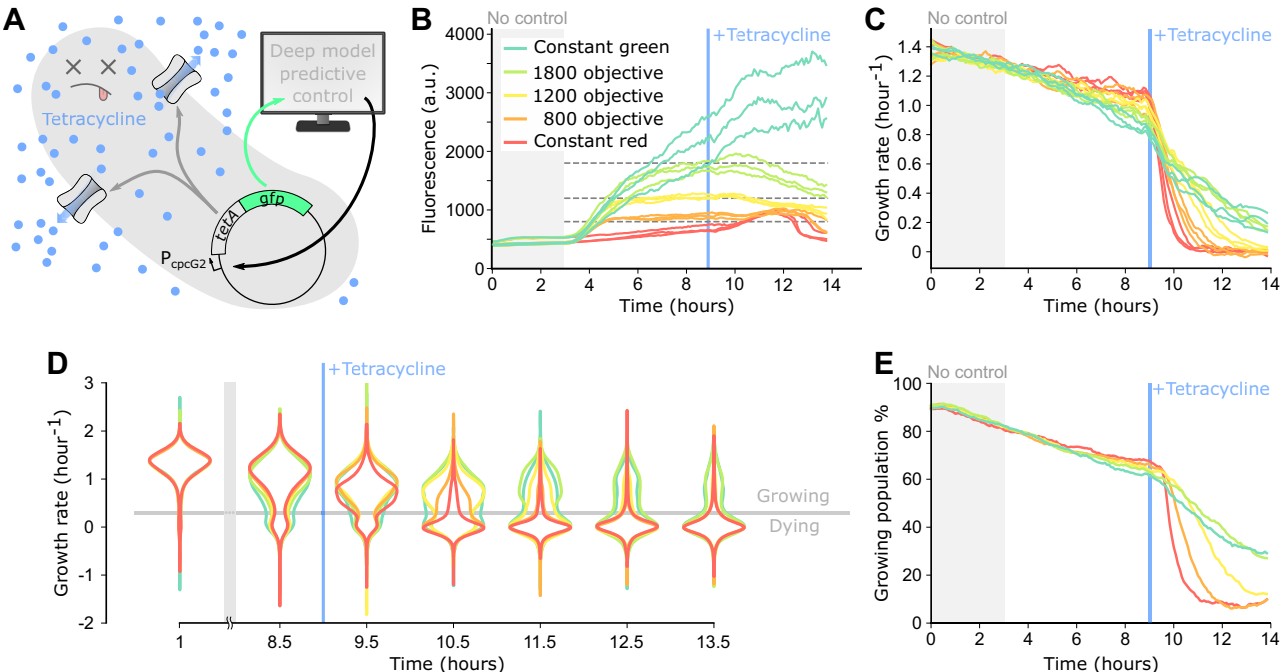

**Fig. 5 | Optogenetic control of the TetA efflux pump drives timing of tetracycline killing in single cells. A** The *tetA* efflux pump gene is placed under control of the $P_{cpcG2}$ promoter for the CcaSR optogenetic system, on an operon upstream of *gfp*. The deep model predictive controller decides on red or green optogenetic stimulations to activate or repress *tetA-gfp* expression based on measured fluorescence. As more TetA efflux pumps are produced, more tetracycline is exported from the cell, preventing cell death. **B** Median fluorescence levels over time for all five control categories across three replicates. The red and green curves correspond to cells exposed exclusively to constant red or green stimulations. The orange, yellow, and yellow-green curves correspond to controlled cells that were randomly assigned to objectives of 800, 1200, and 1800 arbitrary units of fluorescence. Each experiment was conducted with three replicates. For each category and each replicate ~400 cells were controlled, for a total of *n* = ~1200 cells per control category. The objective levels are indicated with dashed gray lines. At t = 9 h, 40 µg/mL of tetracycline was added to the media. **C** Median growth rates for single cells from (**B**). Each curve was smoothed with a Savitzky-Golay filter (length = 15, order = 2) to improve visualization. **D** Growth rate violin plots for all five control categories over time. Data from all replicates were merged for each category. Single-cell growth rates were smoothed with a median filter sliding over a one-hour time window. Horizontal gray line represents the 0.3 h⁻¹ "growing" vs. "dying" growth rate threshold. For visual clarity, the top and bottom 0.1% of the growth rate distributions were filtered out. **E** Percentage of cells in the "growing" sub-population over the total population of cells over time for each control category. Data from all replicates were merged. Single-cell growth rates were smoothed as described in (**D**).

"grainy" result. To quantify performance, we calculated various error metrics over time. After the control starts at t = 3 h, each cell rapidly catches up to its objective, and by t = 6 h the error reaches a minimum (Fig. 4B, Fig. S14B). Following this, the error level increases over time. This degradation in quality is visible in the reconstructed movie, where by t = 32 h most details are barely identifiable. The highest and lowest objective values tend to lead to higher error (Fig. S14C). Rapid shifts in objective value also lead to higher error (Fig. 4C). Interestingly, rapid negative changes appear to be harder to track than rapid positive ones, likely because of the intrinsic dynamics of our biomolecular system where reductions in GFP signal are due to cell division and dilution and not active degradation. In addition, over time a key contributor to the deterioration in control quality appears to be cell aging, as mother cells trapped at the dead end of the mother machine chamber accumulate damage asymmetrically[30,32,37,38]. Indeed, as the experiment goes on, the median growth rate of the population declines, and an increasing number of cells appear to filament (Fig. 4D). The distribution of growth rates throughout the experiment shows that a portion of the cells stop growing altogether (Fig. 4E), and these lower growth rates are correlated with higher control error (Fig. 4F). We note that curation to remove these dead or unhealthy cells would be straightforward given heuristics based on cell length or growth rate, however we have elected to show the full results for all cells here for completeness. More surprisingly, cells in the top 5% of growth rates also exhibit significantly worse control performance (Fig. 4F). To understand the reasons behind this, we compared performance for low and high fluorescence objectives. With low control objectives we observed very little error even at the highest growth rates, while fast growing cells struggled to achieve the high objectives. Higher objectives are likely harder to reach and maintain for faster growing cells because of higher protein dilution rates, and a complex interplay appears to exist between growth rate and control performance.

Our approach using deep model prediction for single-cell control can drive cells to a broad range of dynamic behaviors. Importantly, the speed of the deep learning-enabled control strategy provides an unprecedented level of throughput. Furthermore, this approach provides exquisite accuracy, allowing individual cells to track complex dynamic signals. This opens the door for precise, high-throughput studies of genetic systems.

## Control of antibiotic resistance with TetA

Our initial tests focused on dynamic control of GFP, however our approach should in principle be generalizable to the control of any gene paired with a fluorescent reporter, such that a wide variety of physiological processes could be controlled. To investigate this idea, we made the tetracycline resistance gene *tetA* light-responsive by placing it downstream of the $P_{cpcG2}$ promoter of the CcaSR system, followed by an additional ribosome binding site and the *gfp* gene (Fig. 5A). With TetA and GFP transcriptionally co-expressed, we wanted to evaluate whether fluorescence could be used as a proxy to precisely control TetA levels with our platform. TetA is well-suited as a test case for precision control because it is an efflux pump specific to tetracycline antibiotics[39], and while efflux pumps are beneficial under chemical stresses, they can otherwise be detrimental to cell health if

over-expressed[37,40]. This trade-off can further complicate control, as TetA impacts cell fluorescence levels by modulating growth and therefore dilution rates. Thus, control of TetA provides a realistic test case for linking gene expression in single cells to function.

To test how robust and adaptable our overall approach is, we purposely did not adjust the control algorithm or optimize strain designs. Retraining the timeseries forecasting neural network would likely lead to better control accuracy, however we made the deliberate decision to evaluate the performance of our controller without acquiring strain-specific training data. We also purposefully did not engineer several different variants of this genetic circuit, for example by using different ribosome binding sites. Indeed, with *gfp* in the second position in the operon, we found that the fluorescence signal was weaker than in our experiments with *gfp* control alone, but we only adapted our imaging and optogenetic stimulation parameters. These changes had a small impact on throughput, limiting it to control of ~2,000 cells in parallel per experiment, but required minimal effort with no genetic or model-based alterations.

We performed time-lapse experiments in which cells either received constant stimulations with red or green light or were subjected to deep model predictive control (Fig. 5B, Fig. S15). We randomly assigned cells to five different populations across experiments: In the two constant light populations, we continuously subjected the cells to either red or green light stimulations only, thus showing the full dynamic range of our strain. In the other three closed-loop populations, cells were dynamically controlled to intermediate fluorescence set points at 800, 1200, or 1800 arbitrary units of fluorescence. As before, we began applying deep model predictive control at t = 3 h. At t = 9 h, we added 40 μg/mL tetracycline to the media, perturbing growth and eventually leading to death in susceptible cells. After the introduction of tetracycline, control quality for the entire population progressively deteriorated, as expected given tetracycline's effects on cell health. Notably however, prior to tetracycline introduction, closed-loop control resulted in tight distributions across replicates near the pre-defined control objective, highlighting a major benefit of deep model predictive control for probing subtle links between gene expression and phenotypes. The ability to control this new strain without retraining the timeseries forecasting model and within a tight fluorescence range is also remarkable, and raises the possibility of a modular controller where other genes of interest could be inserted alongside *gfp*.

Next, we asked how set levels of TetA impacted cell growth and survival before and after tetracycline introduction. Several recent studies have revealed that the effect of antibiotic efflux pumps can be heterogeneous at the single-cell level and fluctuate over time[37,40,41], but their effects on cellular physiology and single-cell antibiotic resistance levels can be subtle and may be drowned out by biochemical noise. Our ability to precisely set gene expression in more than 6000 cells allowed us to compare growth rates across graded levels of TetA (Fig. 5C). For all cases, the median growth rate of each category of cells was very similar across replicate experiments, especially for the three closed-loop control cases. Median growth rate generally decayed over time for cells in all populations, but this decay was faster for populations with higher fluorescence, which is consistent with the detrimental effect of TetA expression in the absence of antibiotic. After tetracycline was added to the media however, the situation reversed: while growth rates still decayed for all conditions, higher TetA expression levels led to a slower decrease. Interestingly, not all median growth rates decayed to the same level, indicating potential long-term survival of a subset of the higher-expressing cells following tetracycline treatment.

This led us to investigate single cell effects further, asking how median growth rates relate to population distributions. A decay in median growth rate could indicate that all cells progressively and uniformly slow their growth under tetracycline exposure; alternatively,

it could be the result of two sub-populations, consisting of either living or dead cells, where living cells maintain a constant growth rate until they die and suddenly cease growth. A faster decay in median growth rate would thus be due to a higher number of cells transitioning from the living to dead sub-populations. To investigate this, we looked at single cell growth distributions for the five different experimental categories at different timepoints (Fig. 5D). Up until tetracycline addition, bimodal distributions slowly arose as a small, but increasing, portion of the cells stopped growing. After antibiotic introduction however, growth rapidly halted for nearly all cells that were exposed to constant red light, while for other populations this bimodality was maintained, with higher TetA expression leading to higher resilience (Fig. 5D, Movie S4).

We then partitioned the populations between healthy "growing" and "dying" cells based on a heuristic threshold growth rate of $0.3\,h^{-1}$ (Fig. S16, S17) and quantified the fraction of growing cells in each population (Fig. 5E). For the two lowest TetA expression levels, cells died rapidly and at the end of the experiment only ~8% of cells remained in the healthy sub-population. In contrast, ~30% of cells for the two highest expression levels were able to maintain relatively high growth rates through the end of the experiment (Fig. S18). Taken together, these results indicate that when exposed to tetracycline, cells expressing higher levels of TetA are able to maintain a constant growth rate for longer before they reach a breaking point and stop growing altogether, potentially allowing for stress response pathways to be activated or for beneficial mutations to arise in the interval[40,42].

More broadly, these results demonstrate how deep model predictive control can be used to drive a gene of interest with high precision in single cells. The increase in throughput provided by this approach allowed us to observe the impact on single-cell physiology both temporally and as population distributions. The cell growth burden imposed by TetA expression prior to tetracycline addition is subtle, but our approach clearly illustrates the gradual relationship and trade-off between expression levels and growth rates. Additionally, the dynamic single-cell data reveal insights that cannot be observed from the population level alone, such as the gradual movement of individual cells from healthy to dying sub-populations over time.

## Discussion

Computer-based feedback control of gene expression can be used to study cell dynamics with unprecedented precision[16–18,20,21,43–45]. However, single-cell approaches have historically been limited to the control of a few dozen cells in parallel due to technical limitations associated with cell imaging and the computation times required for algorithms controlling stochastic processes. In this study we leverage recent advances in the fields of machine learning and control theory to improve this throughput by at least two orders of magnitude compared to previous studies[20,21], resulting in real-time control of thousands of cells in parallel. We first showed that neural networks can be trained to predict gene expression in single cells with high accuracy. Then we showed that these prediction networks can be used in a model predictive control framework to precisely drive gene expression dynamics over time-varying and arbitrary objectives without using expert knowledge of the system. Finally, we demonstrated the generality and usefulness of this system by controlling the expression of the resistance gene *tetA* at physiological levels in thousands of cells, without any changes to our hardware, forecasting models, or underlying control algorithm. This allowed us to acquire a large and detailed dataset of cell growth and survival before and after introduction of tetracycline, and to analyze the dynamics of expression burden and stress survival. By imposing precise dynamics onto thousands of cells, our experiments were much more informative than if we had relied on natural gene fluctuations or on crude actuation without feedback control, revealing how subtle expression changes can have a critical impact on cell fate.

While our approach drastically improves throughput and accessibility of real-time control of gene expression, there are limitations that could be addressed with future studies. First, deep neural networks are traditionally seen as "black box" models that are difficult to interpret. However, recent studies in physics show that interpretable mathematical models can be derived from trained networks[46,47]. Another limitation is that training deep learning models requires large amounts of diverse data. This is attainable with single-cell optogenetics, however further work is needed to identify how adaptable our approach would be to other setups, such as microfluidics-based chemical actuation or control of cell populations in bioreactors. The mother machine microfluidic setup itself, which traps a single aging cell for extended periods of time and exposes it to cumulative damage from microscopy imaging, limits how generalizable physiological conclusions can be. Luckily, the increase in throughput permitted by our approach opens the door to model predictive control in alternative experimental configurations such as populations of thousands of cells growing as a monolayer[17,18,32,48]. Another direction to explore regarding aging would be to re-train prediction models on-the-fly, as data is acquired during control experiments, to see if the controller can adapt to changes in system behavior. Yet another limitation is that our predictive model also does not provide any information about prediction uncertainty or noise, which could help improve control accuracy, and may be critical for the control of multistable genetic systems[49,50]. Finally, our use of a model predictive control framework requires that hundreds or thousands of model inferences are made per cell and timepoint. Other data-driven approaches based on imitation learning or reinforcement learning[51] could further improve throughput, as in such frameworks the model would directly infer the optogenetic stimulation to apply to the cell.

Finally, many interesting extensions to our work are possible. A straightforward example would be to apply the same methodology to other genes and compare the impact of fluctuating expression dynamics on phenotypic outcomes such as cell growth or survival. Further, with an additional fluorescent reporter, we could also quantify signal propagation between related genes, or even map out and model gene regulation networks. Because our approach does not make any a priori assumptions about the system to control, it should also be straightforward to apply it to different optogenetic systems besides CcaSR or organisms besides *E. coli*. While the CcaSR system is a powerful and well characterized tool for optogenetic actuation, its activation and inhibition spectra cover a wide range of the visible light spectrum, and its dynamics can be slow compared to post-translational optogenetic systems[15,16,52,53]. Adapting our methodology to other systems could further pave the way towards orthogonal, multiple-input multiple-output optogenetic control. Beyond investigating natural cell processes, our platform could also be used for synthetic biology and metabolic engineering applications[54]. Others have already used similar cell-machine interfaces to simulate the impact of different genetic circuit topologies[18], as a test bench to characterize gene circuit responses[29], or to control methionine metabolism[43]. We envision that this control approach can dramatically expand the scale of these types of studies. The length and accuracy of control experiments may also be improved by utilizing microfluidic devices other than the mother machine, where cell populations can be refreshed by younger progeny and the deleterious effects of cell aging and death can be mitigated. Beyond control of gene expression, neural networks can be used to optimize experiment automation. Researchers have already proposed online optimal experimental design approaches, where experimental inputs are optimized to maximize the information acquired about bacterial growth[55]. In another study, a neural network was used to detect important cellular events and adapt acquisition parameters accordingly[56]. As machine learning based methods for control and automation in applications such as autonomous driving, games, or robotics[51] keep improving, we expect real-time interfaces between these algorithms and live cells to open entirely new ways to conduct biological research.

## Methods

### Plasmids and strains

All experiments use *E. coli* MG1655. For experiments where only GFP was controlled, the *fliC* flagellar gene was deleted to prevent cells from swimming out of the microfluidic traps. The deletion was performed using the Datsenko-Wanner chromosome engineering protocol[57]. We then transformed this strain with plasmids pNO286-3 and pSR58.6 from the CcaSR v3 optogenetic system[28]. For tetracycline resistance experiments, the *fliC* gene was kept, as the deletion altered the physiological response to the antibiotic. We inserted the *tetA* gene from plasmid pRGD-TcR (Addgene #74110) upstream of sfGFP in a transcriptional fusion. The primers GCATTTTTAAcgcagtcaggcac and TCTCCTCTTTtcaggtcgaggtgg were used to amplify the *tetA* gene and the primers ctcgacctgaAAAGAGGAGAAATACTAGATG and cctgactgcgTTAAAAATGCGATCCTAAC were used to amplify the pSR58.6 backbone. The insert and backbone were joined via Gibson assembly[58] and co-transformed with plasmid pNO286-3 in MG1655. This plasmid is available on AddGene.

### Cell cultures and growth media

Before starting experiments, cells were grown overnight in 5 mL LB supplemented with 0.2 g/L Pluronic F-127, 25 μg/ml chloramphenicol, and 50 μg/ml spectinomycin. Pluronic prevents cell adhesion within the microfluidic chip; chloramphenicol and spectinomycin are required for plasmid maintenance. In the morning, cultures were refreshed at a 1:100 ratio in 5 mL of the same media supplemented with 1 g/L of glucose and grown for 3 h. Cells were loaded into the chip as described below. The media used in the microfluidic chip after loading the cells was LB broth supplemented with 0.2 g/L Pluronic F-127 and 1 g/L of glucose. For experiments where only GFP was controlled, we added 5 μg/ml chloramphenicol and no spectinomycin. The reason for these lowered doses of antibiotics is that once loaded in individual chambers the bacteria do not compete with each other, and plasmid loss is less problematic. Chloramphenicol was still used to prevent contamination of the media. For tetracycline resistance experiments, we added no antibiotics to minimize the potential for spurious cross-resistance effects.

### Microfluidics

The master molds for the mother machine microfluidic chips were made with SU-8 resin on silicon wafers using photolithography, based on the design in https://gitlab.com/dunloplab/mother_machine. To produce the microfluidic devices, we poured polydimethylsiloxane (Dow Corning Sylgard 184) and cured it overnight at 75 °C, cut out the chips and punched the inlets and outlets, and removed debris on the chip with Scotch tape. We then bound the chips to a 24 × 50 × 1.5 mm glass slide (Fisherbrand 12-544-EP) after activating both surfaces for 10 s at 100 W in a plasma oven (EMS Quorum 1050X). We found that carefully optimizing plasma parameters and controlling humidity were critical for good binding. Before using the plasma, we used an electronics duster to remove most debris on the glass slide. The chip features 8 parallel channels where media flows, each with 1000 growth chambers of 25 × 1.8 × 1.2 μm dimension.

Before loading the cells, water supplemented with 0.5 g/L Pluronic F-107 was flowed manually through the chip at high pressure. This improved chamber loading rates significantly, allowing >95% of chambers to be filled with cells. Then the cell cultures were spun down at 16,000 g, the supernatant was removed, and the cell pellet was resuspended into ~50 μL media. This high concentration resuspension was then flowed into the chip, and the chip was spun in a tabletop centrifuge at 4696 g to introduce the cells into the mother machine chambers. Finally, the microfluidic chip was connected to a peristaltic pump (Ismatec EW-78001-12) flowing growth media and set up on the microscope. The chip was left on the microscope stage for 3–4 h under constant red light without any imaging to allow the cells to recover

before an experiment began. For tetracycline resistance experiments, the bottle of media was manually swapped at t = 9 h to a new one supplemented with 40 µg/mL tetracycline.

## Microscopy

We used a Nikon Ti2 chassis widefield epi-fluorescence microscope. All imaging was performed with a 100X oil objective (Nikon MRD31905). Both phase contrast and GFP imaging were performed with an eGFP filter cube in the light path (Chroma 49002) to increase acquisition throughput. The fluorescence excitation white light source was a Lumencor SOLA light engine. Image acquisitions were performed with an Andor Zyla 4.2P-USB3 camera. For most experiments, fluorescence light intensity was set to 5%, and exposure was set to 85 ms. For tetracycline resistance experiments, these values were 30% and 100 ms respectively to match the dynamic range of the original strain. We placed an RGB LED ring around the microscope condenser (Adafruit Neopixel 1586) to apply constant illumination to the cells. The cells could thus be exposed to red light (620 nm wavelength) or green light (525 nm wavelength). During the 3–4 h recovery and equilibration period before an experiment was started, cells were exposed to constant red light with this LED ring. A Digital Micromirror Device (Mightex Polygon400) was connected to the illumination light path of the Ti2 chassis to dynamically project patterned images onto the field of view. This allowed us to target specific chambers with red or green light. The light source of the DMD was an X-Cite XLED1 featuring an RDX red LED unit (660–675 nm) and a BGX green LED unit (505–545 nm). In experiments where only GFP was controlled, we applied only red DMD stimulations (60 ms exposure, 2066 mW/cm$^2$) while the LED ring supplied constant green light for the entire experiment. DMD illuminations were the most time-consuming operations in our experimental loop, and constantly shining green light onto the cells with the LED ring allowed us to avoid single-cell green stimulations with the DMD. In tetracycline resistance experiments, cells were targeted with both red (60 ms exposure, 2066 mW/cm$^2$) and green (100 ms exposure, 331 mW/cm$^2$) stimulations while the LED ring was kept red for the entire experiment. These settings allowed us to have a tighter off state. Finally, we used an Arduino Uno microcontroller to coordinate hardware synchronization between the camera, the light sources, the LED ring, and the DMD to increase acquisition speed (see GitLab repository).

All equipment was connected to a workstation computer (HP Z840 with 128 GB DDR4 physical memory, 2 Intel Xeon E5-2623 v4 CPU featuring 4 physical, 8 logical cores each at 2.60 GHz, and an nVidia Quadro P4000 GPU featuring 8GB GDDR5, 1792 CUDA cores at 1.2 GHz) and was interfaced with the Micro-Manager microscope control software v2.0.1[59], and its core API was accessed in Python via Pycro-Manager v0.14[60]. This allowed us to develop Python modules and scripts for highly customized acquisitions and to exploit the Tensorflow v2.6 deep learning library with GPU acceleration. The code we used to control our platform is available on GitLab: https://gitlab.com/dunloplab/pycromanager

## Error metrics

We use two main metrics to assess accuracy between a set of ground truth or control objective values, and a set of model predictions or controlled cell fluorescence values. We represent the objective or ground truth values as $g_t^{(n)}$ and the prediction or fluorescence values as $f_t^{(n)}$, with $t \in [\![1,T]\!]$ as the time index, and $n \in [\![1, N]\!]$ as the sample or cell index. The root mean square error computed across time, for each cell n is:

$$\mathrm{RMSE}_{\mathrm{time}}(n) = \sqrt{\frac{1}{T}\sum_{t=1}^{T}\left(f_t^{(n)} - g_t^{(n)}\right)^2}$$

This metric allows us evaluate the error for a single prediction or a single controlled cell over a period of time, making it possible to look at error as a distribution and to extract median error and percentiles. In some cases we also use the root mean square error computed across cells, at each point in time:

$$\mathrm{RMSE}_{\mathrm{cells}}(t) = \sqrt{\frac{1}{N}\sum_{n=1}^{N}\left(f_t^{(n)} - g_t^{(n)}\right)^2}$$

This metric makes it possible to look at the evolution of error over time, across all cells or samples.

## Open-loop time-lapse experiments and data analysis

Training and validation experiments were performed across 125–150 different locations in the microfluidic chip, each one of them with 27–28 chambers in the field of view (amounting to 3375–4200 mother cells per experiment). Data were acquired in series of 25 positions, where first phase contrast and GFP images were acquired and then optogenetic stimulations for all 25 positions were applied. Phase contrast and GFP acquisitions were both performed using 85 ms exposure times. The red DMD stimulations were performed using 60 ms exposure times. All positions were acquired and stimulated every 5 min. All acquired images were immediately saved to disk. In parallel, the phase contrast images were cropped into smaller images around each chamber in the field of view, and were segmented with our time-lapse analysis software DeLTA v2.0.5 8ceb01[31,32]. Because we were only interested in following the mother cell trapped at the dead end of the chamber, we did not perform DeLTA's tracking step and simply retrieved data corresponding to the cell at the top of the image.

After segmentation, single-cell features were extracted on-the-fly and recorded in a data array. The first of these features is the sequence of optogenetic stimulations applied to the cell, stored as 0 for red stimulation and 1 for green. After image analysis, the mother cell average GFP intensity and cell area (in pixels, 1 pixel ≈4400 nm$^2$) were computed using the mother cell's segmentation mask. We also recorded the average fluorescence of all cells in the chamber, the standard deviation of those levels, and the total number of cells in the chamber. The optogenetic stimulations for a chamber's immediate neighbors were also compiled. Finally, we extracted image sharpness, defined as the mean value of the Laplacian of the cropped phase contrast image of the chamber. In total, 8 single-cell feature timeseries were extracted on-the-fly. These arrays were saved to disk at the end of each experiment. These single-cell feature arrays were then normalized to the [0, 1] range to train and be used as inputs to the neural networks (Supplementary Text). For subsequent feedback experiments, the single-cell feature arrays were normalized to the [0, 1] range on-the-fly using the same functions and parameters.

We performed acquisition experiments for four training and three validation sets. These experiments were performed without feedback control, although features were extracted on-the-fly. The training and validation experiments include a total of 15,898 and 13,811 mother cells, respectively. Each experiment lasted between 16 and 24 h. Optogenetic stimulations were pre-determined as random binary sequences. To ensure that cells would be subjected to long periods with or without DMD stimulations, these sequences were computed by binarizing a one-dimensional random walk (Supplementary Text).

Growth rate was not computed on-the-fly, and was instead computed a posteriori from the cell area data. First, we set area values to NaN ("not a number" in floating point format) for cells and timepoints where cell area or fluorescence levels were too small to represent a cell (less than 0.44 µm$^2$ and 100 a.u. respectively), which are indicative of image analysis artefacts that become more frequent as cells die. Then, at every time point $t$ we computed the growth rate for each cell as:

$$\text{growth}(t) = \frac{\text{area}(t+1) - \text{area}(t)}{\text{area}(t)}$$

Finally, growth values below $-2.4\ \text{h}^{-1}$ were filtered to NaN to avoid time points where divisions occur. While these values could be computed from lineage tracking information, this approach simplifies analysis. To compute population medians or smoothed timeseries, NaN values were ignored.

## Timeseries forecasting

The deep learning models are written and run with the Tensorflow/ Keras library v2.6. They are structured as encoder-decoder networks. The encoder consists of two LSTM[27] layers of 64 and 16 units respectively, that reduce the 8-dimensional hours-long timeseries into a 32-dimensional latent space vector. This encoded representation of the cell's past is then concatenated with the binary vector of potential future optogenetic stimulations. This concatenated vector is the input of the fluorescence prediction decoder. The decoder consists of 5 densely-connected layers[27] of 32 units with rectified linear activation, and a final densely-connected layer of 12, 24, 36, or 48 units depending on the prediction horizon, with a linear activation function. These hyperparameters were selected after evaluating their performance both in terms of prediction accuracy and inference time (Fig. S1).

For training, a random cell and timepoint were picked among the thousands of 8-dimensional, 16–24 h long single-cell trajectories. The cell's trajectory prior to the selected timepoint was extracted as training inputs for a random past period of time ranging between 3 and 12 h (36–144 time points). The cell's optogenetic stimulations and fluorescence for the 1 to 4 hours (12 to 48 time points) following the selected timepoint were also extracted, as training input and ground truth, respectively. We used Keras's built-in mean squared error between the model's predictions and the ground truth as training loss, and used the Adam algorithm[34] for gradient descent with a learning rate of $10^{-3}$.

First, to evaluate prediction performance, models were trained on only the training datasets for 500 epochs, 200 steps per epoch, and 100 samples per batch. We generated validation samples to evaluate model performance during training: At the end of each training epoch, the $\text{RMSE}_{\text{time}}(n)$ was evaluated over 10,000 random samples from the validation dataset (Fig. S19). A potential point of concern was that our limit of 500 training epochs may have been premature, as recent studies have shown that validation loss can drop dramatically, long after model performance appears to plateau[61]. To test this, we also trained the model for 10,000 epochs and observed that the model did not generalize any further (Fig. S20). We found that validation error rapidly plateaus and tends to increase again after 200 epochs while the training loss generally keeps decreasing, pointing to potential overfitting of the network beyond the 200 epoch point. Therefore, we concluded that 200 epochs was the optimal hyperparameter under our training conditions. Finally, based on this conclusion we trained new models over the combined training and validation datasets for 200 epochs, and those models were used for feedback control.

For the linear forecasting model, we provide a detailed explanation for our implementation in Supplementary Text. Briefly, because a linear regression model is mathematically equivalent to a single-layer perceptron with a linear activation function, we re-used the training procedure described above with only minor modifications. For the ODE-based model from Chait et al. [20], we implemented the same ODE model as in that study, fit it to our data, and implemented the same hybrid Kalman filter for state estimation. Details can be found in Supplementary Text, and we provide our implementation with the rest of our code (see Code Availability Statement).

## Deep model predictive control

Once trained, the model was split into its encoder and decoder parts to perform feedback control. The encoder is only run once per input

timeseries, returning the 32-dimensional latent space representation of the cell's past. This representation is then concatenated with potential future control strategies, and fed into the decoder to predict the effect each candidate strategy will have on the cell's future gene expression level. Splitting the model allowed us to run the LSTM encoder only once per cell, which is the slower part of our model since recurrent neural networks are less adapted to parallelization. The model was not re-trained on-the-fly during control experiments.

The deep model predictive control algorithm works as follows: The root mean square error between the output of the prediction decoder and the control objective is computed, and a binary particle swarm optimizer with 40 particles iterates over it 25 times to refine the control strategy (Fig. S9, Supplementary Text). The strategy that is predicted to bring the cell's fluorescence level closest, in terms of $\text{RMSE}_{\text{time}}(n)$, to the pre-determined control objective is selected. We did not constrain strategy selection; any binary sequence of red or green stimulations can be selected by the controller. The first time point of the optogenetic stimulation strategy is applied, and 5 minutes later after the images are acquired and analyzed, the whole process is repeated. See Supplementary Text for a more detailed problem statement.

Although both image segmentation and the control algorithm are fast when run separately on our microscope's computer, constantly switching between deep learning models on the same computer creates significant overhead that severely slows down the execution of both processes. To circumvent this problem, we implemented a small TCP/IP server that runs the control algorithm on a separate laptop (Dell XPS15 9560 with 32 GB DDR4 physical memory, 1 Intel i7-7700HQ CPU featuring 4 physical, 8 logical cores at 2.80 GHz, and an nVidia GeForce GTX 1050 GPU featuring 4GB GDDR5, 640 CUDA cores at 1.34 GHz) instead of the microscope's computer. After image analysis is run on the microscope's computer, the extracted data are sent to this server, and the control strategies are sent back. In future implementations, it is possible that this could be addressed by installing a second GPU on the microscope's computer.

For the deep model predictive control experiment where all cells were assigned the same sinewave control objective, we controlled 2068 cells independently, where cells were randomly assigned to four groups, under the 1, 2, 3, and 4-hour horizon algorithms. For all subsequent experiments, the 2-hour horizon controller was used.

For the concentric sinewaves and the *2001: A Space Odyssey* experiments, pixels in the original movie were randomly shuffled and each experiment completed a subset of the 10,000 pixels of the shuffled movie. The movie was de-shuffled afterwards. For the concentric sinewaves movie, we ran 3 experiments of 3462, 4151, and 3478 single cells. The 1091 extra cells in the third experiment were assigned to re-run the worst performing trajectories of the previous two experiments, and the data from the re-run trajectory was used in the movie over the original. For the *2001: A Space Odyssey* movie, we ran 3 experiments of 3456, 4143, and 3463 single cells, again using the extra 1062 cells to re-run the worst-performing pixels.

## Reporting summary

Further information on research design is available in the Nature Portfolio Reporting Summary linked to this article.

# Data availability

All datasets, processed experimental data, and trained models have been deposited and are available in the Zenodo database under accession code 8114649, https://zenodo.org/record/8114649.

# Code availability

All code used to train neural networks, perform deep model predictive control, analyze data, and plot the figures in this study is on GitLab, https://gitlab.com/dunloplab/deepcellcontrol.

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

## Acknowledgements
We thank Dr. Heidi Klumpe and Dr. Mo Khalil for their helpful comments on the manuscript. This work was supported by NSF grant 2032357 and NIH grant R01AI102922. CMB received support from the NSF Graduate Research Fellowship under grant DGE–1840990.

## Author contributions
J.-B.L. and M.J.D. conceived and designed the study. J.-B.L. implemented the experimental platform and performed all experiments with help from C.M.B. J.-B.L. performed data analysis, developed and fit all prediction models, and implemented the control framework. C.M.B. engineered the antibiotic resistance strain. J.-B.L., C.M.B., and M.J.D. wrote the manuscript.

## Competing interests
The authors declare no competing interests.
