## [Peer Review File · Nature Communications]

Reviewers' Comments:

Reviewer #1:

Remarks to the Author:

The authors have done an excellent job in addressing our comments. We recommend the revised paper for publication in *Nature Communications*.

Reviewer #2:

Remarks to the Author:

I thank the authors for their diligent efforts in addressing my comments. The paper, as it stands, is well presented, articulating the ideas with clarity, and I believe it is fit for publication in *Nature Communications*.

While the paper is comprehensive, I offer two minor suggestions that could further refine and enhance it:

1. I commend the authors for presenting a performance comparison of various forecasting models applied to their specific setting. Such a comparison fortifies their claims about the superior performance of their approach relative to the existing state-of-the-art. Nonetheless, I have the following observation. While the LSTM-based model's median accuracy is evidently superior, it would be insightful to understand how the error distribution differs across the modeling strategies. This will bolster the reader's confidence in the controller's safety and offer a more comprehensive perspective than the isolated examples provided in the supplementary figures. Therefore I suggest the inclusion of a plot that displays the cumulative error distribution of the forecasting models, arranging each experiment from the most to the least accurate.

2. The authors have effectively demonstrated that an LSTM-based model captures the system dynamics more adeptly than both the ODE-based and linear regression-based models. Although I do get that the choice of modeling is the primary distinction between deep MPC and MPC, warranting a comparison of model fitting in isolation from control, I would still recommend showcasing the performance of the closed-loop systems in action.

Point-by-Point Response to Reviewers
Manuscript #: NCOMMS-23-15224-T

Reviewer #1 (Remarks to the Author):

The authors have done an excellent job in addressing our comments. We recommend the revised paper for publication in Nature Communications.

We thank the reviewer again for their insightful comments and questions.

Reviewer #2 (Remarks to the Author):

I thank the authors for their diligent efforts in addressing my comments. The paper, as it stands, is well presented, articulating the ideas with clarity, and I believe it is fit for publication in Nature Communications.

We thank the reviewer for their thorough evaluation of our work, as it drove us to establish the novelty and advantages of our approach more solidly.

While the paper is comprehensive, I offer two minor suggestions that could further refine and enhance it:

1. I commend the authors for presenting a performance comparison of various forecasting models applied to their specific setting. Such a comparison fortifies their claims about the superior performance of their approach relative to the existing state-of-the-art. Nonetheless, I have the following observation. While the LSTM-based model's median accuracy is evidently superior, it would be insightful to understand how the error distribution differs across the modeling strategies. This will bolster the reader's confidence in the controller's safety and offer a more comprehensive perspective than the isolated examples provided in the supplementary figures. Therefore I suggest the inclusion of a plot that displays the cumulative error distribution of the forecasting models, arranging each experiment from the most to the least accurate.

We agree with the reviewer and have added a panel to Fig. S7 that plots the error distributions for the three types of models we evaluated on the validation dataset. The error distributions are all approximately log-normal, indicating that the models do not feature "failure modes" where the prediction becomes particularly bad for a subset of samples.

→ See Fig. S7B

2. The authors have effectively demonstrated that an LSTM-based model captures the system dynamics more adeptly than both the ODE-based and linear regression-based models. Although I do get that the choice of modeling is the primary distinction between deep MPC and MPC, warranting a comparison of model fitting in isolation from control, I would still recommend showcasing the performance of the closed-loop systems in action.

We agree with the reviewer that directly comparing closed-loop performance between the models in the MPC framework would be ideal, but this would represent significant experimental work

and code re-formatting, and the results of the comparison in prediction accuracy and speed already show that using a neural network model leads to a leap in control capabilities (see Fig. 2H-I).